# EMERGENT SO(3)-INVARIANT MOLECULAR REPRESENTATIONS FROM MULTIMODAL ALIGNMENT

## ABSTRACT

Learning molecular representations that are robust to 3D rotations typically requires architectures with built-in symmetry priors or extensive data augmentation. In this work, we investigate whether contrastive multimodal pretraining alone can induce SO(3) invariance in molecular embeddings. We jointly train a continuous 3D-field encoder, based on a vector-quantized generative adversarial network (VQGAN), and a SMILES-based transformer encoder on a dataset of 855,000 molecules, each represented by a DFT-computed electron density grid and a corresponding canonical SMILES string. Both CLIP-style and SigLIP contrastive objectives are used to align representations across modalities. Because SMILES embeddings are invariant to molecular orientation, the contrastive loss implicitly encourages the 3D encoder to produce rotation-consistent representations by aligning different poses of the same molecule to a fixed symbolic anchor. To evaluate geometric generalization, we construct a benchmark comprising 1,000 molecules with five unseen random SO(3) rotations each. The CLIP-based model retrieves at least one rotated variant among its top-10 results for 77% of queries, compared to 9.8% for a unimodal VQGAN baseline, and retrieves three or more variants for 45% of queries (versus 0% baseline). Functional group-wise Recall@10 exceeds 98% for most chemical classes, and clustering by HOMO energy yields a Davies–Bouldin index of 2.35 (versus 34.46 for the baseline), indicating strong chemical organization in the latent space. Additionally, fine-tuning with rotated samples reveals a trade-off between retrieval precision and pose diversity. These results suggest that contrastive multimodal pretraining can yield symmetry-aware molecular representations, even in the absence of explicit equivariant design.

## 1 INTRODUCTION

Learning molecular representations that are both chemically expressive and geometrically invariant remains a central challenge in molecular machine learning Schütt et al. (2018); Satorras et al. (2021). Most 3D molecular models achieve invariance to spatial transformations by explicitly encoding symmetry through architectural design or by leveraging rotation-based data augmentation Thomas et al. (2018); Fuchs et al. (2020); Anderson et al. (2019). These methods assume that symmetry priors must be built into the model to preserve physical consistency, particularly under SO(3) rotations. This raises a fundamental question: *Can pose-invariant representations instead emerge implicitly from the training objective, without enforcing geometric priors through model design?* Chen et al. (2020); Bengio et al. (2013).

We hypothesize that contrastive alignment between invariant symbolic descriptors (e.g., SMILES) and spatially variant 3D fields (e.g., electron densities) can induce pose-consistent molecular embeddings, even in the absence of symmetry-aware architectures Chen et al. (2020); Radford et al. (2021); Wang et al. (2022). This builds on the intuition that multimodal contrastive learning can serve as a *functional regularizer*, promoting semantic alignment across heterogeneous modalities despite differences in spatial representation Xie et al. (2023); Tsai et al. (2019); Robinson et al. (2020).

Multimodal contrastive learning has shown promise in molecular domains by aligning symbolic and topological views of a molecule Wang et al. (2022); Zhang et al. (2024); Kaufman et al. (2024). Nonetheless, existing approaches operate predominantly on graph-based or discrete representations and do not evaluate whether the learned embeddings are robust to arbitrary spatial transformations

Takeda et al. (2023). In particular, it remains unexplored whether contrastive pretraining over unaligned continuous 3D fields can give rise to emergent SO(3) invariance.

In this work, we investigate whether a CLIP-style model trained to align SMILES strings with ab initio-derived 3D electron density grids can learn pose-invariant representations, despite lacking architectural equivariance or rotation augmentation. Our model is pretrained on a dataset of 855,000 molecules, each presented in a canonical orientation, and jointly embeds both symbolic and volumetric views.

To evaluate generalization under spatial transformations, we construct a benchmark of 1,000 molecules, each paired with five randomly rotated SO(3) variants. Our contrastive model retrieves at least one rotated instances in the top-10 for 77.3% of queries, approaching the performance of the SE(3)-equivariant Pos-EGNN baseline (79.1%). Pos-EGNN is a large-scale foundation model trained on 1.4M ab initio simulation snapshots from the Materials Project Trajectory dataset to predict energies, forces, and stress tensors using symmetry-aware message passing ibm.

Beyond retrieval, we probe the latent space for chemical coherence. Without any supervision on quantum properties, the model organizes molecules based on HOMO energies and functional groups: for example, nitrogen-containing species cluster tightly in HOMO-aligned regions. In contrast, the Pos-EGNN latent space—while geometrically grounded—exhibits weaker clustering around frontier orbital descriptors, suggesting that symbolic anchoring plays a critical role in inducing chemically meaningful structure. This organization is quantified by a Davies–Bouldin index of 2.35, compared to 34.46 for a unimodal 3D baseline and 5.53 for the SE(3)-equivariant Pos-EGNN model, indicating superior alignment between geometry and electronic structure.

These findings demonstrate that multimodal contrastive pretraining can induce symmetry-aware molecular representations through emergent behavior, without hard-coded inductive biases. While our approach assumes rotational equivalence across poses—an idealization that may not hold in stereochemically sensitive tasks—it offers a flexible and scalable alternative to equivariant modeling. All code and pretrained models are available at: `https://anonymous.4open.science/r/anonymous-B0BB/README.md`.

## 2 RELATED WORK

Learning molecular representations that incorporate 3D structure has been a longstanding objective in machine learning for chemistry. Early approaches relied on graph-based models augmented with spatial features Schütt et al. (2017); Gilmer et al. (2017), while more recent methods leverage equivariant neural networks Thomas et al. (2018); Anderson et al. (2019); Luo et al. (2025); Satorras et al. (2021). These architectures enforce rotational and translational symmetry by design, often using group convolutions or tensor representations. Although effective, these methods hard-code geometric priors into the model, which may limit flexibility across tasks where symmetries are not strictly preserved.

Beyond equivariance, several works explore data-driven approaches to learning molecular 3D structure. Models such as GemNet Klicpera et al. (2021) and DimeNet++ Gasteiger et al. (2020); Zhu et al. (2023) use angle and distance information explicitly, while diffusion-based models Hoogeboom et al. (2022); Xu et al. (2022) attempt to generate 3D conformers in a probabilistic manner. These methods assume access to accurate conformations or focus on generating new 3D geometries, rather than studying robustness to transformations applied to known structures.

Multimodal learning in molecular domains has focused largely on combining symbolic and graph-based modalities Xiao et al. (2024); Li et al. (2025); Liu et al. (2024). Works such as MolCLR Wang et al. (2022) and Smiclr Pinheiro et al. (2022) demonstrate that contrastive pretraining over graphs or SMILES can improve downstream property prediction. AMOLE Lee et al. (2024) applies a CLIP-style objective to graphs and text but does not incorporate continuous 3D field-based inputs. As a result, existing multimodal methods primarily operate over discrete structural abstractions, limiting their capacity to exploit fine-grained geometric information available in physical electron density fields.

Invariance learning without explicit symmetry enforcement has been explored in vision Chen et al. (2021); Kim et al. (2022), where models trained without augmentations nonetheless exhibit partial

viewpoint robustness. In molecular machine learning, such emergent invariance remains largely unexplored, with most models enforcing rotational symmetry by design Thomas et al. (2018); Satorras et al. (2021). Recent works on SO(3)-equivariant diffusion Hoogeboom et al. (2022) primarily address generative modeling rather than retrieval robustness under unseen transformations.

Our work contributes to this landscape by demonstrating that contrastive multimodal pretraining over symbolic descriptors and continuous 3D grids can induce pose invariance without requiring symmetry-aware architectures. We provide systematic evaluation over rotated benchmarks and relate retrieval stability to chemical and geometric consistency.

## 3 METHODOLOGY

We propose a multimodal contrastive pretraining framework to learn molecular representations that align symbolic descriptors and continuous 3D fields—without relying on symmetry-aware architectural priors. The model jointly embeds SMILES strings and electron density grids derived from ab initio calculations using independent encoders optimized under a contrastive loss. As illustrated in Figure 1, our architecture combines a transformer-based encoder for SMILES with a 3D VQGAN-style convolutional encoder for electron densities. All

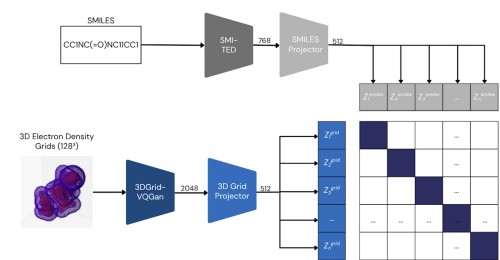

Figure 1: Architecture of the multimodal contrastive model.

parameters—including those from SMI-TED and the 3DGrid-VQGAN encoder—are trained jointly from scratch.

### 3.1 PRETRAINING DATASET

We curate a dataset of 855,000 molecules from PubChem, filtered to include: (i) only main-group elements up to Barium; (ii) a maximum of 30 heavy atoms; (iii) zero net charge; and (iv) no formal charge separation.

Each SMILES string is converted into 50 conformers using RDKit's distance geometry and MMFF94 optimization Landrum (2013). The five lowest-energy conformers are reoptimized using MINDO3 in PySCF Sun et al. (2018), and the conformer with the lowest energy is retained. This structure is further evaluated at the RHF/STO-3G level, and its electron density is projected onto a $128 \times 128 \times 128$ voxel grid, yielding a physically grounded 3D representation without relying on classical graph approximations.

### 3.2 MULTIMODAL CONTRASTIVE PRETRAINING

We align SMILES and 3D electron density representations via contrastive learning. Let $g : \mathcal{X} \to \mathbb{R}^d$ and $h : \mathcal{T} \to \mathbb{R}^d$ denote the 3D and SMILES encoders, respectively. For a batch of $N$ molecule pairs $\{(x_i, t_i)\}_{i=1}^N$, we compute embeddings as $\mathbf{z}_i^{\text{grid}} = \text{Proj}_g(g(x_i))$ and $\mathbf{z}_i^{\text{smiles}} = \text{Proj}_h(h(t_i))$, where Proj denotes a learnable projection head.

**SMILESDFT-CLIP** uses the symmetric InfoNCE loss:

$$\mathcal{L}_{\text{CLIP}} = \frac{1}{2N} \sum_i \left[ \ell(\mathbf{z}_i^{\text{grid}}, \mathbf{z}_i^{\text{smiles}}) + \ell(\mathbf{z}_i^{\text{smiles}}, \mathbf{z}_i^{\text{grid}}) \right],$$

where $\ell(z, z') = -\log \frac{\exp(\text{sim}(z, z')/\tau)}{\sum_j \exp(\text{sim}(z, z'_j)/\tau)}$ and $\text{sim}(z, z')$ is cosine similarity.

**SMILESDFT-SigLIP** employs a sigmoid-based contrastive loss. After normalization $\tilde{\mathbf{z}} = \mathbf{z}/\|\mathbf{z}\|_2$, we define:

$$\text{logits}_{ij} = \exp(\tau) \cdot \langle \tilde{\mathbf{z}}_i^{\text{grid}}, \tilde{\mathbf{z}}_j^{\text{smiles}} \rangle + b, \quad \mathcal{L}_{\text{SigLIP}} = -\frac{1}{N} \sum_{i,j} \log \sigma(y_{ij} \cdot \text{logits}_{ij}),$$

where $\sigma$ is the sigmoid function and $y_{ij} = 1$ for positive pairs, $-1$ otherwise.

### 3.3 3D ELECTRON DENSITY ENCODER

We use a 3DGrid-VQGAN adapted for volumetric inputs to encode electron density grids ibm. The encoder $E(\cdot)$ maps $G$ to a latent tensor:

$$z_e(G) \in \mathbb{R}^{\frac{H}{s} \times \frac{W}{s} \times \frac{D}{s} \times k}, \quad s = 4, \quad k = 512.$$

Latents are quantized using a learned codebook $\{e_j\}_{j=1}^{16384}$:

$$z_q(G) = e_{k^*}, \quad \text{where } k^* = \arg\min_j \|z_e(G) - e_j\|_2.$$

The 3DGrid-VQGAN is trained with:

$$\mathcal{L}_{\text{VQGAN}} = \mathcal{L}_{\text{rec}} + \beta \mathcal{L}_{\text{commit}} + \gamma \mathcal{L}_{\text{adv}},$$

where $\mathcal{L}_{\text{rec}}$ is an $L_1$ reconstruction loss, $\mathcal{L}_{\text{commit}}$ encourages codebook usage, and $\mathcal{L}_{\text{adv}}$ is a 3D PatchGAN adversarial loss. During contrastive training, we use the encoder output *before quantization* and fine-tune all encoder parameters jointly.

### 3.4 SMILES ENCODER

The SMILES modality is encoded using SMI-TED$_{289\text{M}}$ ibm, a pretrained transformer encoder trained on 91 million canonical SMILES strings. Input tokens $X \in \mathbb{R}^{D \times L}$ are processed via RoFormer-style attention:

$$\text{Attention}_m(Q, K, V) = \frac{\sum_{n=1}^{N} \langle \varphi(R_m q_m), \varphi(R_n k_n) \rangle v_n}{\sum_{n=1}^{N} \langle \varphi(R_m q_m), \varphi(R_n k_n) \rangle},$$

where $R_m$ is a position-specific rotation matrix and $\varphi(\cdot)$ is a Fourier feature mapping. A pooled embedding is computed as:

$$\mathbf{z} = \text{LayerNorm} \left( \text{GELU}(X \mathbf{W}_1 + \mathbf{b}_1) \right) \mathbf{W}_2.$$

Unlike prior work, we fine-tune the SMI-TED encoder during contrastive learning, which we find improves performance in both retrieval and structure–property clustering.

### 3.5 TRAINING DETAILS

We train using AdamW with batch size 128 and learning rate $3 \times 10^{-4}$, employing a linear warmup over 1,000 steps. Models are trained for 50,000 steps using both CLIP and SigLIP objectives, with checkpoints selected by retrieval accuracy on a held-out validation set. All experiments are conducted on 4 NVIDIA A100 GPUs.

## 4 EXPERIMENTAL SETUP

We conduct a comprehensive evaluation to assess the extent to which our multimodal model exhibits geometric generalization, chemical organization, and transferability. Our evaluation protocol includes retrieval under both canonical and unseen SO(3) rotations, unsupervised structure–property clustering, and molecular property prediction on the QM9 benchmark.

**Retrieval under** SO(3) **Rotations.** We evaluate retrieval performance in two settings:

1. **Canonical retrieval** – Each query is matched against a corpus of unrotated (canonical) molecules.

2. **Unseen rotation retrieval** – Each query is matched against five rotated spatial variants of each molecule, not observed during training.

To generate unseen rotations, we apply random rigid-body transformations to the atomic coordinates of each molecule. Rotation axes are sampled from the set $\{(0, 0, 1), (0, 1, 0), (1, 0, 0), (1, 1, 0), (1, 0, 1), (0, 1, 1)\}$, and rotation angles are drawn uniformly from $[0°, 360°]$. For each rotated conformer, we recompute the electron density using the same RHF/STO-3G procedure as used during pretraining, ensuring physically valid volumetric fields.

Table 1: Evaluation metrics used to assess geometric and chemical generalization.

| Metric | Description |
|---|---|
| Accuracy@10 | Proportion of queries retrieving the correct molecule within the top-10 results. |
| Recall@10 | Fraction of retrieved molecules belonging to the same functional group. |
| Group-wise Recall@10 | Recall@10 computed for six chemical classes: amines, aromatics, ethers, ketones, halides, and carboxylic acids. |
| Pose diversity | Mean number of distinct rotational variants retrieved in the top-10. |
| Multi-pose retrieval rate | Proportion of queries retrieving at least three distinct rotated variants among the top-10. |

We report the following metrics as in Table 1:

To benchmark invariance, we compare our model against a unimodal 3D electron density baseline (3DGrid-VQGAN) trained without symbolic alignment. This evaluation probes both instance-level and class-level generalization under unseen spatial transformations.

**Structure–Property Clustering.** We assess whether the latent space reflects chemically meaningful organization by analyzing clustering behavior of molecules with similar frontier orbital properties. In particular, we focus on nitrogen-containing species with high HOMO energies—chemically important due to lone-pair reactivity. We quantify cluster quality using the **Davies–Bouldin (DB) index**, where lower values indicate compact, well-separated clusters. This analysis tests whether the model implicitly learns structure–property relationships without supervision.

**Property Prediction on QM9.** To evaluate transferability to downstream tasks, we train linear regression models on frozen multimodal embeddings to predict 12 molecular properties from the QM9 dataset Wu et al. (2018). The encoders are not fine-tuned, ensuring that performance reflects the intrinsic quality of the pretrained representation. We report **mean absolute error (MAE)** on the standard train/validation/test splits and compare against an equivariant baseline embeddings from Pos-EGNN encoder.

# 5 RESULTS

We evaluate the capacity of our multimodal model to achieve pose-invariant molecular retrieval and chemically consistent embeddings without architectural equivariance. The evaluation is organized along two main axes: retrieval under unseen $SO(3)$ rotations and retrieval consistency across known rotations observed during training. Additional analyses include functional group-specific recall and structural similarity assessments.

## 5.1 RETRIEVAL UNDER $SO(3)$ ROTATIONS

We evaluate the ability of our multimodal model to achieve chemically and geometrically consistent retrieval without architectural symmetry constraints. Our experiments are organized into two main settings: retrieval among canonical poses (known rotations) and retrieval under unseen $SO(3)$ rotations.

**Retrieval among Canonical Poses.** In the first setting, retrieval is performed among unrotated (canonical) molecular conformations. Each molecule is embedded in a fixed pose, and retrieval relies solely on feature similarity without any unseen spatial transformations. This setting tests whether the learned embeddings capture molecular identity and functional similarity under ideal alignment.

Retrieval metrics include Top-$k$ Match Accuracy (the fraction of queries retrieving the exact molecule) and functional group (FG) recall, measuring how many retrieved molecules share dominant chemical groups with the query.

Both SMILESDFT-CLIP and SMILESDFT-SigLIP achieve high retrieval performance. At Top-10, SMILESDFT-CLIP achieves 98.8% accuracy, while SMILESDFT-SigLIP reaches 97.6%. The average number of functional group matches within the Top-10 retrieved molecules exceeds eight for both models, demonstrating chemically aligned latent organization.

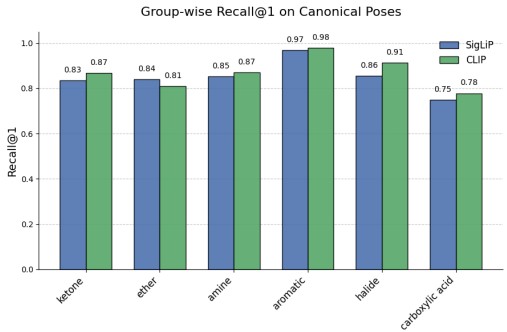

Figure 2: Group-wise Recall@1 for canonical retrieval.

| Model | Top-1 | Top-10 | FG Matches (Top-10) |
|---|---|---|---|
| SMILESDFT-SigLIP | $68.9\% \pm 1.96$ | $97.6\% \pm 0.21$ | 8.33 |
| SMILESDFT-CLIP | $71.4\% \pm 0.83$ | $98.8\% \pm 0.14$ | 8.43 |

Table 2: Retrieval performance among canonical poses.

As shown in Figure 2, aromatic systems exhibit the highest recall (97.9% SMILESDFT-CLIP, 96.8% SMILESDFT-SigLIP), consistent with their distinct electronic signatures. Carboxylic acids, by contrast, show lower recall (77.8% SMILESDFT-CLIP, 75.0% SMILESDFT-SigLIP), likely due to their conformational flexibility. Across all groups, SMILESDFT-CLIP consistently outperforms SMILESDFT-SigLIP at Top-1, indicating sharper discrimination from InfoNCE-based alignment.

**Retrieval under** $SO(3)$ **Rotations.** In this experiment, we evaluate the ability of pretrained models to retrieve molecular representations under unseen rigid-body transformations. To simulate $SO(3)$ rotation invariance, molecules are randomly rotated around arbitrary axes, with rotation angles uniformly sampled from $[0°, 360°]$. Retrieval is performed by querying canonical molecules against rotated versions in embedding space, testing whether models generalize across poses without having observed such transformations during training.

Table 3 summarizes the performance across four models using three metrics: (i) Accuracy@10, which captures exact retrieval of a rotated instance; (ii) Recall@10, which measures class-level or functional group recovery; and (iii) the proportion of queries for which three or more rotated variants appear among the top-10 candidates.

Table 3: Retrieval performance under unseen $SO(3)$ rotations. **Equivariant** indicates SE(3)-equivariant models. Accuracy@10 measures instance-level retrieval; Recall@10 captures functional group recovery; final column reports the percentage of queries retrieving *3* distinct rotated variants in the top-10.

| Model | Equiv. | Modality | Acc@10 | Rec@10 | 3 Rot. Retrieved |
|---|---|---|---|---|---|
| **Ours** | | | | | |
| SMILESDFT-CLIP | ✗ | 3D Grids + SMILES | $77.3\% \pm 0.51$ | $98.4\% \pm 0.13$ | $45.3\% \pm 0.57$ |
| SMILESDFT-SigLIP | ✗ | 3D Grids + SMILES | $46.1\% \pm 0.57$ | $98.9\% \pm 0.13$ | $43.0\% \pm 0.63$ |
| SMILESDFT-CLIP (finetuned) | ✗ | 3D Grids + SMILES | $\mathbf{85.4\% \pm 0.42}$ | $\mathbf{99.4\% \pm 0.09}$ | $\mathbf{57.9\% \pm 0.54}$ |
| SMILESDFT-SigLIP (finetuned) | ✗ | 3D Grids + SMILES | $\mathbf{88.4\% \pm 0.37}$ | $\mathbf{99.6\% \pm 0.08}$ | $\mathbf{59.1\% \pm 0.52}$ |
| **Baselines** | | | | | |
| Pos-EGNN | ✓ | Atom Positions (SE(3)) | $79.1\% \pm 0.44$ | $99.2\% \pm 0.12$ | $51.2\% \pm 0.51$ |
| 3DGrid-VQGAN | ✗ | 3D Grids Only | $9.1\% \pm 0.22$ | $2.3\% \pm 0.02$ | $0.0\% \pm 0.01$ |

Table 3 reports retrieval performance under unseen $SO(3)$ rotations, comparing our multimodal models to both equivariant and non-equivariant baselines. Fine-tuned variants of SMILESDFT-CLIP and SMILESDFT-SigLIP—trained on 1,000 randomly selected molecules with five randomly rotated poses each—achieve the highest Accuracy@10 (85.4% and 88.4%, respectively), outperforming the SE(3)-equivariant Pos-EGNN baseline (79.1%) despite lacking explicit symmetry priors. All multimodal models exhibit strong functional group recovery (Recall@10), with fine-tuned versions reaching 99.6% (SMILESDFT-SigLIP). Furthermore, over 57% of fine-tuned model queries retrieve at least three distinct rotated variants in the top-10, exceeding the equivariant baseline (51.2%) and substantially outperforming the unimodal 3DGrid-VQGAN model, which fails under rotation. These

results suggest that contrastive multimodal pretraining, when exposed to a modest set of diverse poses, can induce rotation-consistent representations without requiring architectural equivariance.

This result underscores the central contribution of our approach: emergent rotational invariance arises from multimodal contrastive pretraining, even in the absence of architectural equivariance or rotation augmentation. The SMILES representation remains invariant under rotation and acts as a semantic anchor. Minimizing the contrastive loss aligns the spatially-variant 3D electron density fields with these invariant anchors, inducing consistent embeddings across different orientations. Pooling operations further reduce sensitivity to local spatial deformations, contributing to pose-robust representations.

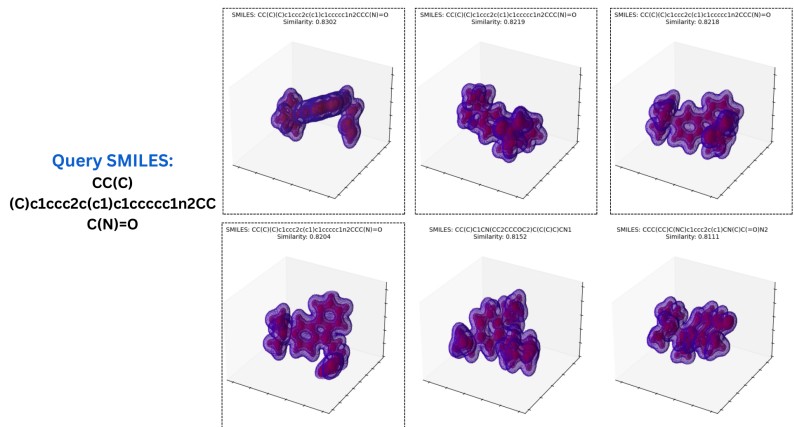

Figure 3: Visualization of retrieval results under unseen rotations using SMILESDFT-CLIP. Query SMILES: `CC(C)(C)c1ccc2c(c1)c1ccccc1n2CCC(N)=O`. Retrieved electron density grids are matched with corresponding SMILES and cosine similarity scores. The model retrieves four perfect matches and one close structural analogue, illustrating robustness to SO(3) transformations.

Figure 3 illustrates a retrieval example using SMILESDFT-CLIP. Among the six closest retrieved samples, four are exact matches under distinct rotations, and one is a structurally similar analogue. This highlights the model's ability to capture both spatial and semantic consistency.

Group-wise Recall@10 scores (Table 4) reveal high retrieval robustness across functional groups. Both multimodal models achieve near-perfect recovery for aromatic and ketone-containing compounds. Slightly lower recall for carboxylic acids may stem from their conformational flexibility and smaller spatial extent in grid representation, which challenges invariant matching.

Table 4: Recall@10 across functional groups under unseen SO(3) rotations.

| Functional Group | SMILESDFT-CLIP | SMILESDFT-SigLIP |
|---|---|---|
| Amine | 0.987 | **0.994** |
| Aromatic | 0.999 | **1.000** |
| Ether | **0.987** | 0.981 |
| Ketone | 0.987 | **1.000** |
| Halide | 0.961 | **0.978** |
| Carboxylic Acid | **0.893** | 0.890 |

In summary, our results show that contrastive multimodal pretraining can induce pose-invariant molecular representations without relying on symmetry-aware inductive biases. By leveraging the invariant nature of symbolic descriptors during alignment, the model internalizes spatial consistency across orientations. This emergent behavior bridges the gap between architectural equivariance and semantic invariance, opening new directions for building chemically robust models from weak supervision alone.

## 5.2 STRUCTURE–PROPERTY RELATIONSHIP

To evaluate whether the learned latent representations reflect chemically meaningful structure–property relationships, we analyze clustering behavior based on the HOMO (Highest Occupied Molecular Orbital) energy, a key descriptor of molecular reactivity. Nitrogen-containing species are of particular interest due to the strong influence of nitrogen lone pairs, which elevate HOMO energy and enhance molecular reactivity.

In the QM9 dataset, nitrogen-containing molecules comprise only 9.10% of the total population but represent 32.81% of the top decile in HOMO energy. Capturing such functional and electronic trends in the learned embedding space—without direct supervision on quantum properties—is a critical test of the model's chemical fidelity.

To quantify clustering quality, we compute the Davies–Bouldin (DB) index, which penalizes overlapping or diffuse clusters (lower is better). Table 5 summarizes the DB scores across models. Notably, the SMILESDFT-CLIP-based multimodal model achieves the lowest DB index (2.35), indicating a tightly organized latent space with well-separated clusters that align with HOMO energy variations. In contrast, the position-equivariant Pos-EGNN model, despite its architectural symmetry priors, yields a higher DB index (5.53), suggesting weaker alignment with electronic structure. This is surprising, as equivariant models are expected to encode physically grounded representations, but lack symbolic anchoring to enforce chemical alignment.

Table 5: Davies–Bouldin (DB) index for structure–property clustering by HOMO energy (lower is better). SMILESDFT-CLIP achieves the lowest Davies–Bouldin index, indicating tight HOMO-aligned clustering. Symbolic input (SMILES) plays a critical role in structuring latent space, even in the absence of equivariant design.

| Model | SMILES | 3D Grids / Atom Positions | DB Index |
|---|---|---|---|
| **SMILESDFT-CLIP** | ✓ | 3D Grids | **2.35** |
| SMI-TED | ✓ | ✗ | 2.82 |
| MoLFormer | ✓ | ✗ | 4.28 |
| Pos-EGNN | ✗ | Atom Positions (SE(3)) | 5.53 |
| 3DGrid-VQGAN | ✗ | 3D Grids | 34.46 |

Figures 4 visualize 2D projections of the learned latent spaces, with colors representing HOMO energy and triangle markers highlighting nitrogen-containing species. The SMILESDFT-CLIP latent space reveals compact clusters strongly correlated with HOMO energy and clearly segregated nitrogen-rich regions, supporting the hypothesis that contrastive multimodal pretraining promotes chemically meaningful representation learning.

In contrast, Pos-EGNN—despite encoding atom positions in an equivariant manner—produces a more diffuse and intermixed embedding space, with nitrogen-containing species scattered across regions of varying energy. This suggests that architectural symmetry alone does not guarantee property-aligned representations unless supported by complementary semantic signals. VQGAN and SMI-TED provide further contrast: the former shows disorganized embeddings due to lack of symbolic anchoring, while the latter partially captures structure–property alignment, but lacks geometric context.

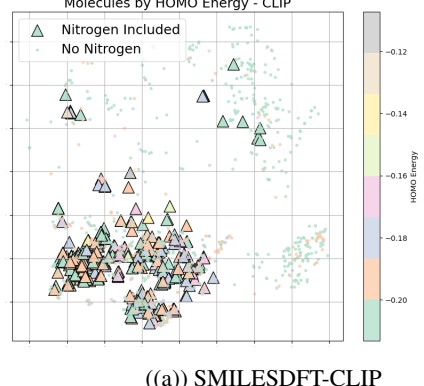 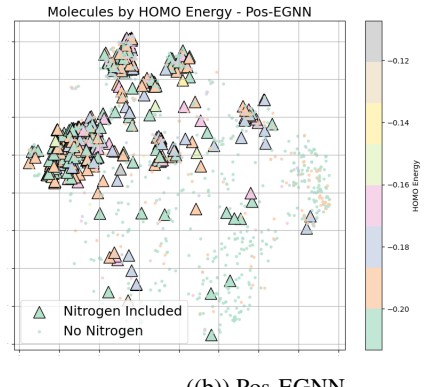

((a)) SMILESDFT-CLIP     ((b)) Pos-EGNN

Figure 4: Latent space projections colored by HOMO energy. Triangular markers denote nitrogen-containing molecules. SMILESDFT-CLIP shows compact, chemically coherent clusters with strong alignment to HOMO energy and nitrogen enrichment. Pos-EGNN yields more diffuse structure, despite its SE(3)-equivariance.

These findings emphasize that contrastive multimodal learning acts not merely as a cross-modal alignment strategy, but as a *functional regularizer* that filters and reinforces task-relevant structural patterns. The invariant SMILES anchor encourages the 3D encoder to focus on chemical features consistent across orientations, facilitating the emergence of rotationally robust, semantically grounded embeddings. Importantly, the superior performance of SMILESDFT-CLIP over Pos-EGNN challenges the assumption that architectural equivariance alone is sufficient for property-aware representation learning, and points to the power of symbolic supervision in organizing chemical space.

## 5.3 Property Prediction on QM9

We assess the downstream utility of our pretrained representations on the QM9 benchmark, which comprises 12 regression tasks spanning electronic, thermodynamic, and geometric properties. Mean absolute error (MAE) is reported in QM9-standard units. Using a pre-trained linear probe setup, we evaluate SMILESDFT-CLIP and SMILESDFT-SigLIP without task-specific fine-tuning to isolate representation quality. As shown in Table 6, both SMILESDFT-CLIP and SMILESDFT-SigLIP consistently outperform the SE(3)-equivariant Pos-EGNN baseline across most tasks. SMILESDFT-CLIP achieves the lowest MAE on 8 of 12 properties—including $\epsilon_{\text{HOMO}}$, $C_v$, and $\langle R^2 \rangle$—while SigLIP is competitive, especially on thermodynamic targets ($U$, $U_0$, $H$, $G$)

Table 6: Mean Absolute Error (MAE) on QM9 regression tasks. All models are evaluated in a frozen linear probe setting. **Blue** and **Orange** highlight the best and second-best results, respectively.

| Category | Property (Unit) | Pos-EGNN (Equivariant) | SMILESDFT-CLIP (Non-equivariant) | SMILESDFT-SigLIP (Non-equivariant) |
|---|---|---|---|---|
| **Electronic** | HOMO energy $\epsilon_{\text{HOMO}}$ (eV) | 0.0093 | 0.0083 | 0.0090 |
| | LUMO energy $\epsilon_{\text{LUMO}}$ (eV) | 0.0141 | 0.0110 | 0.0118 |
| | Energy gap (eV) | 0.0165 | 0.0135 | 0.0144 |
| | Dipole moment $\mu$ (Debye) | 0.6288 | 0.6836 | 0.7243 |
| **Thermodynamic** | Internal energy $U$ (eV) | 6.8596 | 2.8141 | 2.3437 |
| | Internal energy at 0K $U_0$ (eV) | 6.8308 | 2.8403 | 2.3316 |
| | Enthalpy $H$ (eV) | 6.8503 | 2.8137 | 2.3402 |
| | Free energy $G$ (eV) | 6.8335 | 2.8098 | 2.3706 |
| **Geometric** | Heat capacity $C_v$ (cal/mol·K) | 0.6622 | 0.4450 | 0.4455 |
| | Polarizability $\alpha$ (bohr$^3$) | 1.5346 | 1.0771 | 1.1791 |
| | Spatial extent $\langle R^2 \rangle$ (bohr$^2$) | 70.5140 | 45.2012 | 46.5561 |
| | ZPVE (eV) | 0.0064 | 0.0038 | 0.0040 |

Despite lacking symmetry-aware priors, both models outperform Pos-EGNN on geometry-sensitive metrics such as polarizability ($\alpha$) and spatial extent ($\langle R^2 \rangle$), suggesting that contrastive symbolic alignment can induce symmetry-consistent behavior through emergent structure in the latent space.

## 6 Limitations

Our approach assumes that rigid SO(3) rotations preserve molecular semantics, which may not generalize to stereochemically sensitive or highly flexible molecules. The use of RHF/STO-3G electron densities, while providing quantum-consistent inputs, adds computational cost and may limit scalability. Although both encoders are fine-tuned jointly, the model is not explicitly trained to align different conformers of the same molecule. Future extensions could incorporate conformer-invariant objectives, lightweight electronic representations, or hybrid training with physicochemical supervision.

## 7 Conclusion

We show that contrastive multimodal pretraining between SMILES and 3D electron densities yields chemically meaningful, pose-invariant representations—without symmetry-aware architectures or rotation augmentation. The model generalizes across SO(3) rotations and reflects orbital energy structure, despite no geometric or quantum supervision. Symbolic anchoring emerges as a simple but effective inductive signal. Future work may explore conformer-aware or property-specific extensions.

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
