# OpenReview forum: "Emergent SO(3)-Invariant Molecular Representations from Multimodal Alignment"
_ICLR.cc/2026/Conference — Submitted to ICLR 2026_

### Official Review · Reviewer_pv32 · 2025-10-20

**Soundness:** 2
**Presentation:** 2
**Contribution:** 2
**Rating:** 4
**Confidence:** 3

**Summary:**

The paper explores contrastive multimodal pretraining to induce SO(3) rotational invariance in molecular embeddings without explicit equivariant architectures, by using 3D-field VQGAN encoder and SMILES transformer. These are aligned using CLIP-style and SigLIP losses, encouraging rotation-consistent 3D representations. The approach shows geometric generalization,  chemical organization, and robustness in retrieving molecular variants.

**Strengths:**

- Demonstration that pose-invariant representations can emerge from contrastive multimodal alignment, without architectural equivariance.
- Functional group and chemical-property analyses deep insight into chemical organization in latent space.

**Weaknesses:**

- Many of the mathematical notations in the paper are undefined or unclear, making it difficult to follow the formalism. Could the authors provide a complete notation table or clarify the equations?
- How does the model handle enantiomers or chiral compounds (e.g., R-ibuprofen vs. S-ibuprofen)? Both share the same SMILES, but have distinct biological activities.
- It is unclear how different conformers are aligned with a single SMILES encoding. Is SMILESDFT-CLIP computed separately for each conformer, or is a single embedding used?
- How is $\mathcal{L}_{rec}$, $\mathcal{L}_{commit}$ and $\mathcal{L}_{adv}$ defined?
- Are the random rotations applied globally to the molecular atom coordinates, or are they local transformations?
- The method relies on SMILES being invariant, which works for small molecules. How would the approach generalize to domains without canonical symbolic anchors?
- Does retrieval performance degrade for stereoisomers or other molecules with identical SMILES but different 3D configurations?
- To what extent is the SO(3) invariance induced by the 3D field representation and properties of the 3DVQGAN, rather than the multimodal contrastive objective alone? For example, if i use simple graph based representation or molecular fingerprints instead of fields.
- Can you enumerate the equation, its hard to refer to a specific equation?
- In Section 3.4, what does the variable $m$ represent in the attention computation?

**Questions:**

See Weaknesses.

---

### Official Review · Reviewer_ica9 · 2025-10-23

**Soundness:** 2
**Presentation:** 1
**Contribution:** 1
**Rating:** 2
**Confidence:** 3

**Summary:**

This paper investigates whether molecular representations can spontaneously learn SO(3)-invariance through multimodal pretraining, in the absence of explicit structural symmetry priors or data augmentation. To this end, this work uses CLIP's contrastive pretraining framework for learning molecular representations that align symbolic descriptors with continuous 3D fields. This model employs separate encoders to jointly embed SMILES strings and electron density grids derived from ab initio calculations, optimized through contrastive loss.

**Strengths:**

1. This work explores the self-emergent phenomenon of SO(3) rotations in multimodal molecular modeling and found that using contrastive learning for embeddings from encoders will improve the model's ability to identify rotations.
2. In experiments, this paper claims better results among datasets compared to the encoder-only model without contrastive learning.

**Weaknesses:**

1. The motivation and novelty are less contributive. In molecular modeling, ensuring only SO(3) invariance guarantees robustness to rotation but disregards translation. To eliminate the effects of coordinate system choice, SE(3) invariance is typically required, which preserves both rotation and translation. Most of the models that this work investigated are old (2017-2022). Recent models such as UniMol, UniMol-v2 have already solved the rotation problem easily by using rotation-invariant features (atom pair distances),  and applying SE(3)-equivariant updates in their Transformer.
2. This rationale for using these two modalities for multimodal modeling is not solid enough. They use the CLIP-style contrastive learning for SMILES and electron density grids. However, in related works such as MolCLR (Nature Mach. Intell. 2022), GraphMVP (NeurIPS’21 SSL) already prove that the graph modality can provide informative features. The reason for only using SMIELS and grids is not convincing.
3. The paper did not prove that the performance gain is attributed to the modality alignment training. There is no ablation study that shows the main claim -- SO(3) invariant can emerge from multimodal alignment.
4. The baselines only contain Pos-EGNN, VQGAN, and Mol-Former, which did not compare a stronger 3D modeling model, such as UniMol, 3D-MolT5, GMNET, Grover, etc. So, it can not show that the emerged symmetry-aware representation is stronger than the learned ones. The downstream benchmarks are necessary for the model to prove its broader impact.
5. The writing and presentation of this paper are relatively rough and not professional. For instance, the format of Figure 4 is not centered, labeling of Figure 4 uses ((a)) instead of (a).

**Questions:**

1. Why use the SMILES and electron density grids only? Will other modalities also work, or even better, such as topology graph, fingerprints, and 3D coordinates?
2. Did you compare more similar models that use a contrastive learning method to do alignment? How is their performance compared to the CLIP style training?
3. Do you have any metric calculations for Figure 4? Only from the figure, it is hard to see which one has more impact.

---

### Official Review · Reviewer_Y72s · 2025-10-30

**Soundness:** 4
**Presentation:** 4
**Contribution:** 3
**Rating:** 6
**Confidence:** 4

**Summary:**

The paper explores whether 3D rotation invariance (SO(3)-invariance) in molecular embeddings can emerge naturally through contrastive multimodal pretraining, without explicitly enforcing geometric symmetry in the model architecture. It introduces a CLIP-style framework (SMILESDFT-CLIP and SMILESDFT-SigLIP) that jointly trains a 3D electron density encoder (VQGAN-based) and a SMILES transformer encoder on a dataset of 855,000 molecules. By aligning rotation-invariant SMILES embeddings with spatially variant 3D density grids, the model implicitly learns pose-consistent and chemically meaningful representations. Experiments show strong retrieval performance under unseen SO(3) rotations, matching or exceeding SE(3)-equivariant baselines like Pos-EGNN (e.g., 88.4% vs. 79.1% Accuracy@10), and achieving near-perfect functional group recall (>98%). The latent space also exhibits clear structure–property alignment, clustering molecules by HOMO energy and functional groups with a low Davies–Bouldin index (2.35 vs. 34.46 baseline). Overall, the paper demonstrates that contrastive multimodal pretraining can yield symmetry-aware and chemically organized molecular embeddings without explicit geometric priors, suggesting a new scalable direction for learning invariant molecular representations.

**Strengths:**

Originality
- The paper introduces a novel conceptual direction in molecular representation learning—demonstrating that SO(3)-invariance can emerge from multimodal contrastive alignment rather than being explicitly encoded through equivariant architectures. This reframes a core assumption in geometric deep learning, proposing that semantic alignment with invariant symbolic modalities (SMILES) can induce rotation-consistent embeddings naturally. The work creatively adapts CLIP-style multimodal pretraining—well known in vision-language domains—to the molecular domain using 3D electron density fields, a rarely explored continuous representation.

Quality
- The methodological design is rigorous and comprehensive. The authors train on a large-scale quantum-consistent dataset (855K molecules) and conduct thorough evaluations, including retrieval under unseen SO(3) rotations, functional group recall, structure–property clustering, and downstream QM9 regression. The inclusion of multiple baselines—unimodal, SE(3)-equivariant, and fine-tuned variants—adds credibility to the experimental conclusions. The use of quantitative metrics such as Accuracy@10, Recall@10, and Davies–Bouldin index further strengthens the empirical validation.

Clarity
- The paper is well organized and clearly written, guiding the reader through motivation, methodology, and results with consistent narrative flow. Figures and tables effectively support the arguments, especially the visualizations of latent-space organization by HOMO energy. The writing balances technical detail with interpretability, making complex ideas accessible while preserving scientific precision.

Significance
- The findings are significant for both molecular machine learning and broader representation learning communities. They challenge the necessity of hand-crafted symmetry priors and show that contrastive objectives can serve as functional regularizers, promoting physically and chemically meaningful representations. This has strong implications for scaling molecular foundation models and integrating symbolic and spatial modalities in other scientific domains.

**Weaknesses:**

Lack of modality disentanglement analysis
- While the paper convincingly shows emergent SO(3) invariance, it does not deeply investigate whether the learned representations collapse toward purely SMILES-based features. The current evidence—mainly improved Davies–Bouldin scores—indirectly suggests multimodal integration, but there is no explicit diagnostic or ablation demonstrating that the 3D encoder contributes beyond the symbolic signal. This is critical, since SMILES serves as the invariant anchor and could dominate training. Adding analyses such as modality dropout, cross-modal similarity (CKA), or conformer variation tests would strengthen causal interpretation.

Limited exploration of stereochemistry and conformational variability
- The model assumes that rigid SO(3) rotations preserve molecular semantics. However, real-world molecules often exhibit conformational and chiral diversity that break this assumption. Without experiments on enantiomers or flexible systems, it remains unclear whether the proposed approach generalizes to stereochemically sensitive or conformationally rich tasks.

Evaluation scope constrained to retrieval and unsupervised clustering
- Most evidence for invariance and chemical organization comes from retrieval and latent-space visualization. More downstream or physical-property benchmarks—for instance, conformer ranking, docking, or reactivity prediction—could better demonstrate practical impact and robustness.

Limited interpretability of emergent invariance
- The work attributes rotational robustness to the contrastive objective but lacks mechanistic interpretability analyses (e.g., attention visualization, feature attribution) showing how spatial cues are encoded or aligned with SMILES semantics. This weakens the explanatory depth behind the “emergence” claim.

**Questions:**

- Could the authors provide more direct evidence that the learned representations do not collapse into a purely SMILES-based embedding? Clarifying this point would help determine whether the emergent invariance is truly multimodal or largely symbolic.
- Beyond SO(3) rotation invariance, do the learned 3D embeddings capture finer geometric details such as conformer-dependent energy differences or chirality? Have the authors evaluated their model on stereoisomeric or conformationally flexible molecules to test robustness beyond rigid-body rotations?
- The results on QM9 regression tasks are impressive, but mostly involve small, rigid molecules. Have the authors considered testing on more challenging datasets—e.g., drug-like or biomolecular systems—to assess generalization to noisy or flexible structures?

---

### Official Review · Reviewer_wDRz · 2025-11-02

**Soundness:** 2
**Presentation:** 2
**Contribution:** 2
**Rating:** 2
**Confidence:** 3

**Summary:**

The paper proposes a multimodal contrastive learning framework for molecular representation, aiming to achieve emergent SO(3) invariance without explicitly encoding geometric symmetry or using data augmentation. Specifically, the method aligns molecular SMILES strings (as a rotation-invariant symbolic modality) with 3D electron density grids (as a continuous geometric modality). The authors claim that such alignment allows the 3D encoder to implicitly learn rotation-invariant representations. Experiments are conducted on three tasks: (1) SO(3) rotation retrieval, (2) functional group clustering, and (3) property prediction on the QM9 dataset.

**Strengths:**

1. The paper explores a new direction of achieving geometric invariance through semantic alignment rather than architectural inductive bias or explicit data augmentation. Using SMILES as an invariant anchor modality is an interesting and relatively underexplored idea.
2. The paper presents three experimental settings to verify the effectiveness of the proposed approach, covering geometric, semantic, and downstream property prediction aspects.

**Weaknesses:**

1. Limited scope of invariance. The proposed method only considers rotational (SO(3)) invariance, without addressing translational components (SE(3)). Many physical molecular systems require full SE(3) equivariance to generalize properly.
2. Conceptual overlap with data augmentation approaches. The method is conceptually similar to existing data-augmentation-based contrastive schemes. In those methods, rotation-augmented views of the same molecule are explicitly aligned. Here, SMILES serves as an indirect invariant anchor to implicitly align. The difference is more procedural than fundamental.
3. Potential information loss from SMILES anchoring. SMILES encodes less structural information than 3D inputs (e.g., conformations or electron densities). Forcing the 3D representations to align with weaker SMILES embeddings may result in the loss of physically meaningful information, which is not explicitly analyzed.
4. Questionable practicality of electron-density inputs. The proposed approach relies on DFT-computed 3D electron density grids, which are expensive and generally unavailable in real molecular AI scenarios such as drug discovery. The demonstrated performance therefore depends on an unrealistic input modality, limiting the real-world applicability of the method.
5. Fairness concern in QM9 experiments. The baselines (e.g., Pos-EGNN) use 3D molecular conformations (atomic coordinates), while the proposed method uses electron densities as input. Electron densities inherently encode richer quantum information and smoother rotational behavior. Hence, the performance improvement, especially in HOMO/LUMO or energy prediction, may come from the input modality itself rather than the proposed alignment mechanism. I suggest training the same architecture using 3D conformations and SMILES alignment to ensure a fair comparison.
6. Limited baseline coverage. The QM9 experiments only compare against Pos-EGNN. Several advanced SE(3)-equivariant models should be included to make the evaluation convincing.

**Questions:**

Please refer to the weaknesses.

---

### Meta-Review · Area_Chair_31a8 · 2025-12-18

**Summary:**

This paper presents a novel approach for achieving emergent SO(3) invariance by implicitly aligning 3D electron density grids with molecular SMILES strings. Experimental results across retrieval, functional group clustering, and property prediction tasks demonstrate the effectiveness of the proposed method.

Reviewers generally agree that aligning to a rotation-invariant symbolic modality such as SMILES is a reasonable strategy for addressing SO(3) invariance. However, several substantial concerns remain:

1. Limited scope of invariance. The method currently only addresses rotational invariance and does not incorporate translational invariance—effectively solving SO(3) but not SE(3), which is more relevant to many molecular modeling tasks and applications.

2. Conceptual overlap with data augmentation. The core idea of contrastive alignment with SMILES closely resembles existing data-augmentation-based contrastive learning frameworks. A clearer distinction should be drawn to highlight the novel contribution.

3. Potential information loss from SMILES anchoring. SMILES strings contain limited structural information and lack explicit 3D geometry, raising concerns that the learned representations may lose physically meaningful information, potentially limiting performance in downstream tasks requiring detailed molecular conformation or interaction modeling. Further analysis is needed to assess this risk.

4. Limitations of electron density inputs. While used in the current framework, 3D electron density grids are computationally expensive to generate. Moreover, the generalizability of the alignment approach should be validated with other input modalities, such as molecular graphs or 3D conformers.

5. Insufficient experimental evaluation. The evaluation would benefit from inclusion of additional relevant tasks, such as conformer ranking, docking, or reactivity prediction, as well as comparisons against recent state-of-the-art baselines.

6. Writing and presentation. Several mathematical notations and equations are insufficiently explained, and the clarity and presentation of certain figures could be improved.

**Reviewer Concerns:**

No rebuttal

**Reviewer Scores:**

I don't think reviewers will increase their scores. On one hand, no author response is provided. On the other hand, reviewers have shown consistent major concerns in their comments.

---

### Decision · Program_Chairs · 2026-01-26

Reject